# Revealing the spatiotemporal complexity of the magnitude distribution and *b*-value during an earthquake sequence

Marcus Herrmann ®[1] ✉, Ester Piegari ®[1] & Warner Marzocchi ®[1]

The Magnitude–Frequency-Distribution (MFD) of earthquakes is typically modeled with the (tapered) Gutenberg–Richter relation. The main parameter of this relation, the *b*-value, controls the relative rate of small and large earthquakes. Resolving spatiotemporal variations of the *b*-value is critical to understanding the earthquake occurrence process and improving earthquake forecasting. However, this variation is not well understood. Here we present remarkable MFD variability during the complex 2016/17 central Italy sequence using a high-resolution earthquake catalog. Isolating seismically active volumes ('clusters') reveals that the MFD differed in nearby clusters, varied or remained constant in time depending on the cluster, and increased in *b*-value in the cluster where the largest earthquake eventually occurred. These findings suggest that the fault system's heterogeneity and complexity influence the MFD. Our findings raise the question "*b*-value of what?": interpreting and using MFD variability needs a spatiotemporal scale that is physically meaningful, like the one proposed here.

Beroza et al.[1] recently highlighted that current earthquake catalogs achieve a high level of detail that likely contains more information about earthquake occurrence, allows testing of existing hypotheses, and potentially improves earthquake forecasting. One of the main ingredients for earthquake forecasting and seismic hazard models is the Magnitude–Frequency-Distribution (MFD) of earthquakes, which carries information about the proportion between small and large earthquakes. The MFD is typically modeled with the Gutenberg–Richter (GR) relation and its *b*-value (the slope of the GR relation), which can be used to infer the occurrence rate of large earthquakes from small ones. The *b*-value is observed to vary in space and time[2–8], which is thought to be primarily related to variations of the stress state in the crust[9–11]. The *b*-value is also considered as an indicator for other conditions in the crust, which are directly or indirectly related to the stress state, such as faulting style[7,12], locked or creeping fault patches[2,4,13,14], material properties[15,16], fluid pore-pressure perturbations[3,6,17,18], and critical nucleation length[19], among others[11] (and references therein). *b*-value variations may therefore have an important role in improving our physical understanding of earthquake occurrence.

Estimating the *b*-value appears trivial in theory (after all, it is simply the rate parameter of an exponential distribution), but not in practice. Several aspects affect the ability to resolve representative *b*-value variations in earthquake catalogs, such as:

1. the quality of the data, its spatiotemporal selection, and the various ways of sampling it[4,20];
2. the sample size and available magnitude range[20–24];
3. the used magnitude scale, magnitude binning, and maximum likelihood estimator[21,22,25,26];
4. the assumption of the underlying MFD model for the upper end (unbounded, tapered, or truncated) and the detection of departures from an exponential-like GR distribution at the lower end (due to the inherent and potentially varying incompleteness)[24–28], i.e., the estimation of the magnitude of completeness, $M_c$, as the lower magnitude threshold.

Although this list is not exhaustive, these considerations highlight that the outcome of a *b*-value analysis highly depends on expert judgment and/or subjective choices. In fact, recent scientific studies[29,30] and

[1]Dipartimento di Scienze della Terra, dell'Ambiente e delle Risorse; Università degli Studi di Napoli 'Federico II', Naples, Italy.
✉e-mail: marcus.herrmann@unina.it

discussions[31,32] reemphasized that choices have to be specific, meaningful, and reproducible to obtain robust results that contribute to a better understanding of the underlying physical processes. It appears that this field of study requires well-defined schemes and analysis steps. Moreover, choices are critical for real-time applications that need to run automatically, e.g., for operational earthquake forecasting (OEF) purposes[33]. Assessing the influence of expert choices and various modeling ideas on the forecasting performance needs community efforts such as the Collaboratory for the Study of Earthquake Predictability (CSEP)[34,35], which tests forecasting models prospectively in a controlled environment.

Here we argue that a complex earthquake sequence with multiple ruptured fault segments can further bias the MFD and $b$-value analysis: If the MFD varies temporally among distinct zones of a fault system, an averaged view over the whole sequence or a smoothed view over a finite scale (either in space or time) will neglect or mask those variations and may lead to inappropriate or biased inferences. Instead, an MFD analysis may become more physically meaningful and less ambiguous when accounting for distinct seismogenic zones and their evolution during the sequence. This rationale is the subject of this study. It has already been shown that the MFD can significantly differ in adjacent but well-defined zones of induced seismicity[36]. Here we focus on a complex sequence and propose to investigate the spatiotemporal behavior of the MFD and $b$-value by spatially isolating the most seismogenic zones and further dividing them temporally.

We use the 2016/17 central Italy (hereafter 'CI2016') sequence as an example due to its complex tectonic structure, cascading evolution, and the availability of high-resolution catalogs. The CI2016 sequence occurred in the central Apennines, one of Italy's most seismically active areas, and was marked by a cascade of three main events[37]: the $M_w$6.0 ($M_L$6.0) Amatrice event on 24 August 2016, the $M_w$5.9 ($M_L$5.8) Visso event on 26 October 2016, and the $M_w$6.5 ($M_L$6.1) Norcia mainshock on 30 October 2016. On 18 January 2017, four $M_w$5.0–5.5 events followed near Campotosto[37]. These seven events have been caused by movements on southwest-dipping normal faults and they ruptured multiple fault segments, activating a complex fault system[38–43].

The CI2016 sequence is particular in that it features seismicity in a ~1 km-thick subhorizontal detachment at around 10 km depth, which intersects with and confines almost the entire normal fault system above[38,40,43–45]. Such a feature was already observed in the Apennines at a depth of 15–20 km[46], which suggested the presence of a buried subhorizontal thrust related to (the deepest part of) the Apennines build-up. It generally appears as a flat layer, and high-resolution catalogs resolved it as a slightly east-dipping, irregular structure (i.e., with locally varying depth and thickness)[38]. This feature was interpreted as a midcrustal shear zone[45], which decouples the upper and lower crust. Prior seismicity was found to mostly occur along this structure[45], suggesting that it was loaded tectonically and eventually favored the unlocking of the shallower faults through stress transfer. Partially overlapping fault fragments were identified in this structure[43].

Magnitude statistics of CI2016 have been investigated in several recent studies using different spatiotemporal scales. Montuori et al.[47] mapped the $b$-value on a 2-km grid using the 80 nearest events; they found that the Amatrice event originated in an area with a high $b$-value and subsequently reduced the $b$-value to the north and south, suggesting a high potential for further large events. Gulia and Wiemer[48] resolved $b$-value changes relative to the background $b$-value both in time using events within focal-mechanism-driven boxes surrounding the Amatrice and Norcia epicenters, and in space on a 2-km grid using the 250 nearest events; they found a $b$-value variation during the course of the sequence, in particular (i) a drop after the Amatrice event (especially in the area to the north where the Norcia mainshock occurred afterward), interpreted as a still impending large earthquake, and (ii) a $b$-value increase after the Norcia mainshock, interpreted as a substantially reduced chance for a further large earthquake similar to

the tectonic background rate. Garća-Hernández et al.[49] used a multi-scale approach to resolve the $b$-value continuously in time and spatially on a grid; they also observed a "marked drop of the $b$-value" after the Amatrice event (resolved spatially and in depth) and a recovery of the $b$-value to the background level after the Norcia mainshock; they could exclude that these variations were caused by an increased $M_c$ after the main events.

In this study, we reanalyze this sequence using a high-resolution catalog and introduce an alternative perspective for studying MFD variability—using a spatiotemporal scale that considers the 3-D distribution of recorded seismicity. A cluster analysis of the sequence using density-based algorithms lets us spatially isolate the most seismogenic zones; temporal periods are defined by the occurrence time of the largest events. We demonstrate that this approach proves beneficial in resolving the spatiotemporal variation of the MFD and $b$-value. For instance, we resolve what happened in the days before the largest event (Norcia) in its associated seismogenic zone. Rather than solely focusing on $b$-value estimates, we consider it important to exploit more information from the MFD, e.g., by assessing and comparing its exponential-like part and reporting the $b$-value stability as function of $M_c$. We show that the MFD behaves in a complex manner among the spatially isolated clusters throughout the sequence. Our findings reflect on the appropriate spatiotemporal scale to resolve the $b$-value and challenge existing approaches.

## Results

### Description of clusters

Using the high-resolution catalog of Tan et al.[50], we spatially isolated the five largest seismogenic zones (Cluster 1–5, hereafter abbreviated with C1, C2, etc.) following the procedure described in Methods. Figure 1 shows that the obtained clusters are not randomly distributed, but instead highlight the complex spatial structure of the sequence. For instance, C1 comprises seismicity in the northern part of the subhorizontal detachment, parts of the normal fault (Mt. Vettore) that ruptured during the Norcia mainshock, and this mainshock hypocenter itself. C2 represents seismicity in the southern part of the subhorizontal detachment, and C3 captures the shallow northern part of the sequence, including the Visso hypocenter. C4 and C5 relate to small-scale structures. These five clusters correspond to the largest volumes of high hypocenter density (Supplementary Fig. 1). The Amatrice event does not belong to any of the main clusters because the area around its hypocenter is devoid of earthquakes[39,40,50]. The Campotosto events were also not assigned to a main cluster.

Figure 2 shows that each cluster has a distinct temporal activity. For instance, C1 was active throughout the sequence until the Campotosto events; C2 was quiet after the Visso event until the Norcia mainshock while C3 was very active in this period. C4 and C5 were mostly active toward the end of the sequence, along with the other clusters in roughly comparable proportions. Supplementary Fig. 2 and Supplementary Note 1 summarize the cluster statistics in terms of size and ratio for each period, making it more apparent that ≥ 50% of the earthquakes in each period belong to a cluster. Moreover, up to two clusters were dominating each period except for the last period.

### Cluster-based MFD analysis using the whole sequence

For the statistical analysis of the MFD, we follow the procedure described in Methods. Table 1 and Fig. 3 indicate differences and similarities in the MFD among the clusters. In particular, Table 1 suggests that C1, C2, and C3 have identical MFD shapes, but that the MFD of C1 and C2 are distinct from the ones of C4 and C5. There is a tendency that C1 differs from C2, although not statistically significant. Figure 3c provides more details about the MFD behavior in terms of the $b$-value as function of $M_c$. For instance, the largest clusters C1–C3 (red, blue, and green, respectively) have comparable $b$-values (-1.2) at their corresponding $M_c^{\text{Lilliefors}}$, but behave differently for increasing $M_c$:

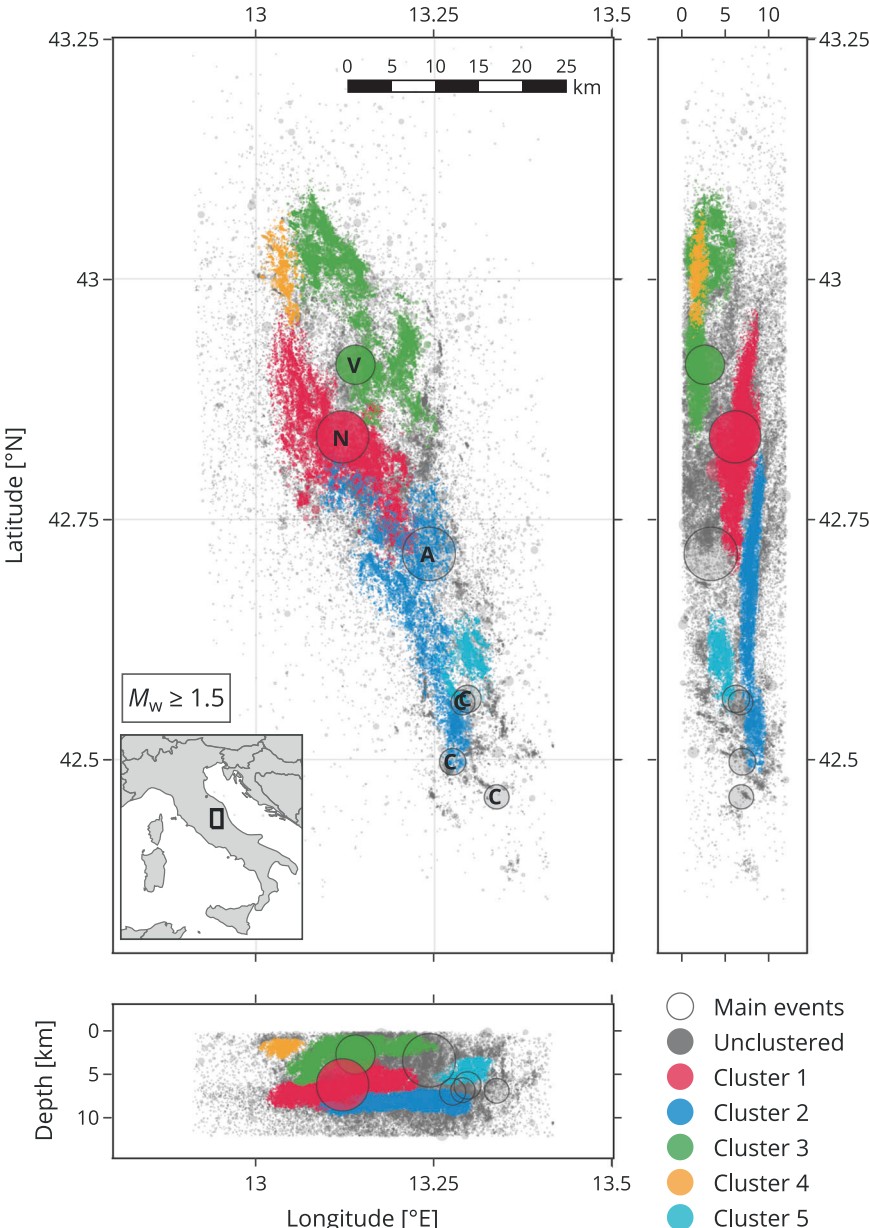

**Fig. 1 | Map view and depth sections of the 2016/17 central Italy ('CI2016') seismicity with identified clusters (see legend).** The depth sections are to scale. To better reveal the structure of the individual clusters, the earthquake hypocenters are plotted ascending by their cluster number on top of 'unclustered' hypocenters, neglecting a physically correct appearance. The main events Amatrice, Visso, Norcia, and four Campotosto events are represented by larger circles; in the map view, they are annotated with the respective initial letter (A, V, N, C). Supplementary Fig. 1 shows the hypocenter density for the same data.

for $M_w \geq 3.0$, the $b$-value is much higher in C3 than in C1 or C2. The small-scale clusters C4 and C5 (yellow and cyan, respectively) show the highest overall $b$-value. The Lilliefors $p$-value (Fig. 3b) is useful to judge the reliability of the $b$-value; a $p$-value dropping below 0.1 indicates that the $b$-value for C1 and C3 below $M_w 2.0$ does not relate to a persistent exponentiality with $M_c$, which can have several reasons (Supplementary Note 2.2) and necessitates an inspection of the MFD in individual periods, as done in the following subsection.

For the sake of completeness, we repeated the analysis using local magnitudes, $M_L$ (Supplementary Fig. 3), which introduces a different MFD behavior for the individual clusters due to a narrower exponential range (Supplementary Note 2.3).

**Cluster-based MFD analysis using temporal subsets**
We extend the spatial analysis by a temporal component using three periods that exclude the short-term aftershock incompleteness (STAI)

between the main events, namely 'pre-Visso', 'pre-Norcia', and 'pre-Campotosto' (see Methods). Table 2 provides a more granular breakdown of MFD variations above $M_c^{\text{Lilliefors}}$ than Table 1, also temporally within the same cluster. For instance, in C1, only pre-Visso and pre-Norcia are distinct; in C2, no period is distinct, and in C3, pre-Visso is distinct from the other two periods. The MFD in pre-Campotosto is never distinct in any cluster. Comparisons among clusters for the same temporal period show no significant differences between C1 and C2, but when comparing C1 or C2 with C3. (Note that the sample size of C2 in pre-Norcia is very small (26 earthquakes), which reduces the power of the KS test to detect potential differences for pairs that include this subset.) The most unique subset is C3 during pre-Visso, which differs from almost all other subsets. Of all 36 pairs, 15 (42%) are significantly different.

Further investigating the MFDs in terms of a $M_c$-dependent $b$-value (Figs. 4 and 5) provides a more nuanced discrimination. The most

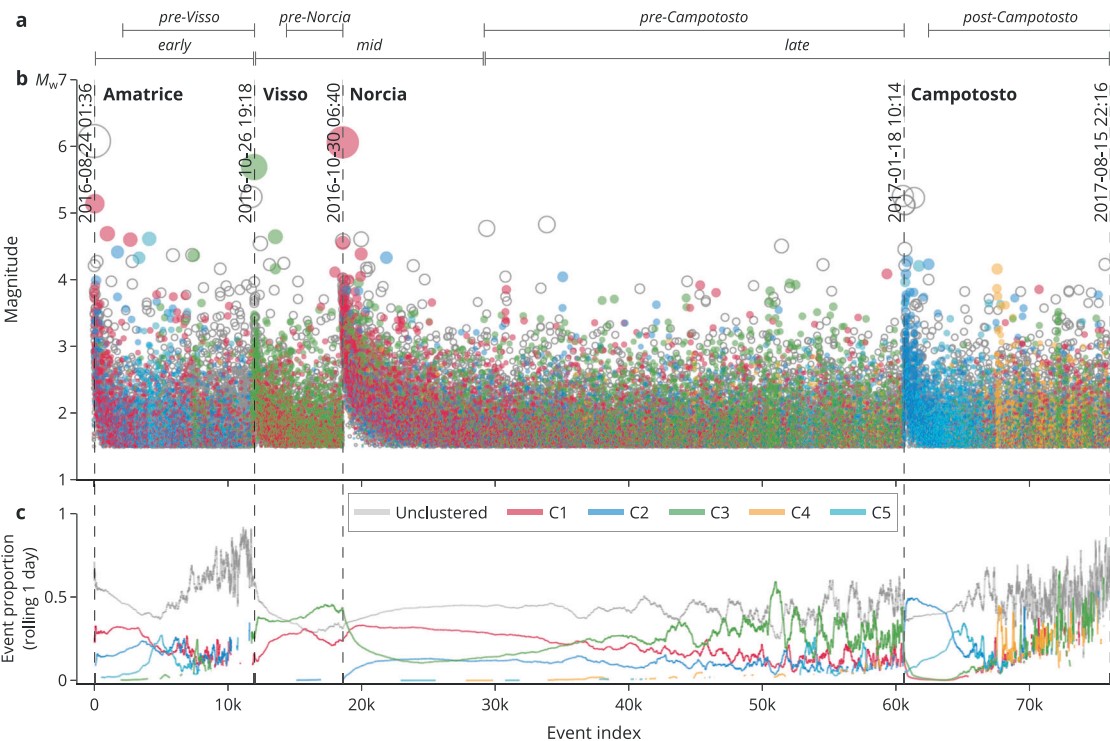

**Fig. 2 | Temporal evolution of CI2016 seismicity colored by cluster association (see legend).** The horizontal axis represents the index of the shown earthquakes. **a** Overview of the temporal subsets. **b** Magnitude vs. index of individual events (empty circles denote 'unclustered'); the occurrence times of main events are annotated vertically. **c** Proportion between the number of earthquakes in each cluster and the total number of earthquakes using a rolling window of the previous 24 h.

remarkable observation is that the *b*-value in C1 is highest before the Norcia mainshock—it has increased after the Visso event from 1.4 to 1.6. After the Norcia mainshock, the *b*-value remained at a high level (1.5 in the pre-Campotosto period). In C2, the *b*-value remained high at ~1.45 both before the Visso event and after the Norcia mainshock. (This cluster does not contain enough earthquakes in the pre-Norcia period to estimate a *b*-value.) In C3, which contains the Visso event, the *b*-value increased from 1.0 in pre-Visso to 1.4 in pre-Norcia, at which level it stayed also after the Norcia mainshock.

Figure 5 facilitates a temporal comparison of the MFD among the clusters. In pre-Visso, the *b*-value is similar in C1 and C2 at around 1.4, and much lower in C3 (1.0). Prior to the Norcia mainshock, the *b*-value increased both in C1 and C3 (to $1.4-1.6$); C2 does not provide enough data. After the Norcia mainshock (i.e., pre-Campotosto), the *b*-value remains elevated in C1–C3 ($1.3 - 1.5$) and C1 and C2 have similar *b*-values again. After the Campotosto events (Supplementary Fig. 8, 'post-Campotosto'), the *b*-value still remains elevated in C1–C3 ($1.4 - 1.5$).

For the sake of completeness, we repeated the analysis using $M_L$ (Supplementary Figs. 4, 5, and Supplementary Note 2.3), which reproduces our main findings qualitatively with comparable relative *b*-

value changes, albeit the *b*-value behaves differently as function of $M_c$ owing to the scale change. For a comparison using temporal periods that include STAI, see Supplementary Note 2.4 and Supplementary Figs. 6–8.

## Discussion

We found that the individual earthquake clusters, which represent the most active zones of this complex sequence, are characterized by a significantly different MFD behavior. In particular, the MFD experienced variations as temporal changes and spatial differences, or remained identical within one cluster throughout the sequence. Finding this complex MFD variability highlights that the spatio-temporal scale to select seismicity and resolve the *b*-value must be physically meaningful. In the following, we first discuss the observed temporal behavior, followed by a discussion of spatial differences and similarities, an interpretation of our findings, and a summary with implications and outlooks.

Regarding the temporal evolution, the most striking observation is the gradual *b*-value increase in the cluster where the strongest earthquake eventually occurred (C1). Apparently, a high *b*-value did not prevent the nucleation of a large rupture in this cluster. This resolved behavior differs from the general observation that the *b*-value decreases prior to large earthquakes[5,48,51–53], albeit similar observations to ours do exist[54]. The increasing *b*-value in C1 after the Visso event highlights that activity in one cluster may influence the MFD in another one. After the Visso event, the *b*-value also increased in its own cluster (C3), which corroborates that large earthquakes may influence the MFD in their surrounding[55]. The later Norcia mainshock, however, did not alter the MFD in the three main clusters further and the *b*-value remained high—also after the Campotosto events, which occurred in the southern part of the sequence. This stagnating *b*-value highlights that the MFD eventually became insensitive to large earthquakes even though it had experienced significant temporal variations in the same

**Table 1 | Pairwise comparison of the cumulative magnitude distribution of each cluster against the others**

|            | Cluster 1 | Cluster 2 | Cluster 3 | Cluster 4 | Cluster 5 |
|------------|-----------|-----------|-----------|-----------|-----------|
| Cluster 1  |           | 0.089     | 0.26      | **2.2e-06** | **1.2e-06** |
| Cluster 2  | 0.089     |           | 0.59      | **0.024**   | **0.0084**  |
| Cluster 3  | 0.26      | 0.59      |           | 0.49        | 0.36        |
| Cluster 4  | **2.2e-06** | **0.024** | 0.49      |             | 0.51        |
| Cluster 5  | **1.2e-06** | **0.0084** | 0.36     | 0.51        |             |

*p*-values of two-sample Kolmogorov–Smirnov tests (Methods). Statistically significant *p*-values are highlighted in bold.

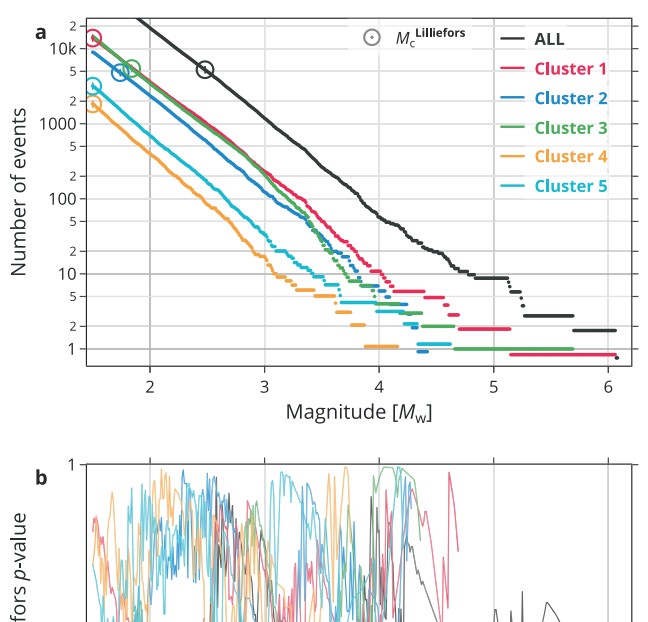

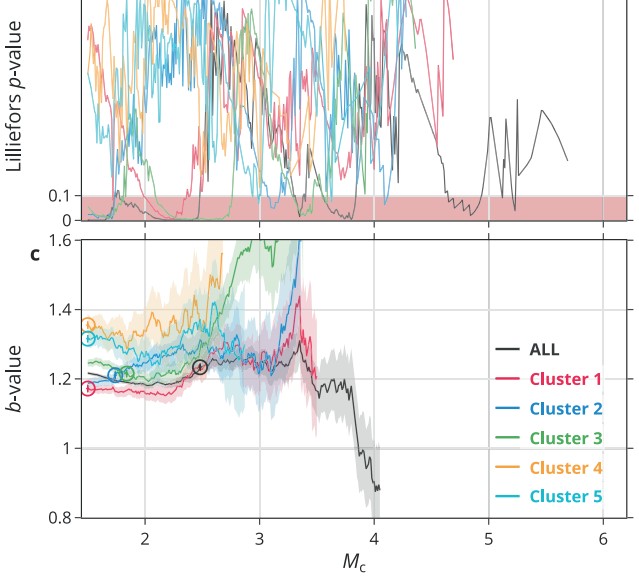

**Fig. 3 | Magnitude statistics for all data (black) and individual clusters (see legend). a** Representing the data in terms of their magnitude–frequency distribution (MFD). Note that a tiny value is added to each MFD (between −0.1 and 0.1) to avoid visual overlaps at large magnitudes. **b** The Lilliefors $p$-value (assuming an exponential distribution as null hypothesis) as a function of lower magnitude cutoff, or magnitude of completeness, $M_c$ (see Methods). **c** The estimated $b$-value (the slope of the fitted Gutenberg–Richter relation, see Methods) as a function of $M_c$ with the shading representing $1\sigma$. The $M_c^{\text{Lilliefors}}$ estimates are indicated for each cluster in **a** and **c** with a circle marker. Supplementary Fig. 3 shows the same analysis using local magnitudes.

seismogenic zones earlier. This ambiguous MFD character is compounded by its behavior in C2, where the MFD locally remained constant throughout the sequence—apparently unaffected by surrounding seismicity.

When comparing clusters spatially regarding the whole sequence, we found differences in the MFD between the largest clusters (C1 and C2) and the smaller ones (C4 and C5). The former have overall lower $b$-value estimates, which are due to the stronger influence of STAI as a result of their proximity to larger earthquakes. In fact, the $b$-value is underestimated in periods that include STAI (Supplementary Note 2.4 and Supplementary Figs. 7–9). Excluding STAI in the individual time periods, we found spatial MFD differences among the largest clusters (C3 differing from C1 and C2). Simultaneously, MFD similarities coexisted among these clusters (C1 and C2), although we do not have evidence for every time period, such as for pre-Norcia when C2 only provides few samples. C1 and C2 have in common that they represent

the majority of seismicity in the subhorizontal detachment at depth (its northern and southern extension, respectively). With their MFD differing from C3 in each individual period and tending toward a higher $b$-value indicates that this subhorizontal detachment is not only tectonically distinct from the shallower normal faults (see Introduction), but also in terms of the MFD.

Although our study focuses on raising awareness of appropriately resolving MFD and $b$-value variations, we briefly speculate about the underlying causes for our most remarkable observations in this sequence. The marked MFD variability among the clusters over time may reflect a heterogeneous stress field and/or a complex fault geometry with significant contributions from the subhorizontal detachment. In other words, the different spatiotemporal behavior of the elastic energy release may be related to some particular (and unknown) features of the geological setting. Moreover, a complex rupture process is suggested by the fact that only some main events belong to clusters—a result of the different hypocenter densities surrounding these events. Assuming the $b$-value is inversely related to the stress state[9,10], the generally higher $b$-value in the subhorizontal detachment could be caused by the structure's reduced capacity to accumulate stress. Instead of accumulating stress, it preferentially transfers stress to the shallow fault system, favoring its unlocking[45]. In fact, this subhorizontal thrust is known to release microearthquakes quasi-continuously along its entire length[38,56], occasionally in minor sequences[57,58], but not hosting larger earthquakes (which should have an extensional mechanism). The very high $b$-value prior and close to the hypocenter of the Norcia mainshock could be explained with (i) the generally high $b$-value in the subhorizontal detachment because C1's pre-Norcia seismicity occurred within its north-eastern extension, whereas its pre-Visso seismicity was located in a shallower part (Supplementary Fig. 10); and (ii) the previous two main events (Amatrice and Visso) and their aftershocks gradually releasing built-up strain and reducing the overall stress in the fault zone. A noteworthy side observation is that the Norcia mainshock nucleated in between the pre-Norcia and the pre-Visso subset of C1 (i.e., the aftershock zones of Amatrice and Visso, respectively, see also Improta et al.[39]), which is consistent with observations that large earthquakes tend to nucleate at the rim of seismic clouds[59,60] and the cascading stress transfer hypothesis[61,62].

In summary, our study demonstrated that the spatiotemporal isolation of seismicity clusters resolves a distinct MFD behavior among the most active zones over time, including influences between them. We therefore argue that the MFD highly depends on the observed seismogenic zone. Since the most seismogenic zones in turn govern the overall MFD behavior of a sequence, a consideration of the activity in individual clusters allows us to decompose and analyze the most important contributions of a complex sequence. Our findings highlight the delicate issue of choosing an appropriate spatiotemporal scale to resolve the $b$-value, challenging existing approaches: A too large scale merges potentially different MFD behavior in individual seismogenic zones and a too fine resolution obscures the tectonic relation and neglects the statistical robustness. The cluster-based approach presented here uses the seismicity distribution itself to choose a scale that is physically meaningful and statistically robust. This strategy may help to reduce the amount of expert judgment and subjective choices, paving the way for replicable MFD analyses and a unified interpretation of MFD and $b$-value variability. For instance, a spatial scale based on density-based clustering can provide appropriate reference volumes (e.g., to determine a background $b$-value for each zone—an analysis that we omitted only due to inconsistent moment magnitude estimates, see Methods). Besides the spatiotemporal scale, other factors and choices complicate a meaningful interpretation; we discuss several of them in Supplementary Note 2, for example MFD exponentiality, STAI, the used magnitude scale, and the used catalog. These

**Table 2 | Pairwise MFD comparison of temporal subsets**

| | Cluster 1 | | | Cluster 2 | | | Cluster 3 | | |
|---|---|---|---|---|---|---|---|---|---|
| | (pre-V.) | (pre-N.) | (pre-C.) | (pre-V.) | (pre-N.) | (pre-C.) | (pre-V.) | (pre-N.) | (pre-C.) |
| C1 (pre-Visso) | | **0.0037** | 0.33 | 0.59 | 0.74 | **0.03** | **2.1e-05** | 0.86 | 0.79 |
| C1 (pre-Norcia) | **0.0037** | | 0.23 | **0.01** | 0.72 | **0.044** | **1.3e-09** | **0.0056** | 0.052 |
| C1 (pre-Campotosto) | 0.33 | 0.23 | | 0.52 | 0.42 | 0.21 | **5.8e-06** | 0.62 | **0.013** |
| C2 (pre-Visso) | 0.59 | **0.01** | 0.52 | | 0.78 | 0.28 | **2.5e-06** | 0.51 | 0.068 |
| C2 (pre-Norcia) | 0.74 | 0.72 | 0.42 | 0.79 | | 0.51 | 0.093 | 0.42 | 0.26 |
| C2 (pre-Campotosto) | **0.03** | **0.044** | 0.21 | 0.28 | 0.5 | | **7.6e-08** | **0.0087** | **0.018** |
| C3 (pre-Visso) | **2.1e-05** | **1.3e-09** | **5.8e-06** | **2.5e-06** | 0.093 | **7.6e-08** | | **3.2e-06** | **0.012** |
| C3 (pre-Norcia) | 0.86 | **0.0056** | 0.62 | 0.51 | 0.42 | **0.0087** | **3.2e-06** | | 0.099 |
| C3 (pre-Campotosto) | 0.79 | 0.052 | **0.013** | 0.068 | 0.26 | **0.018** | **0.012** | 0.099 | |

Statistically significant *p*-values are highlighted in bold.
Like Table 1, but for three periods of Cluster 1, 2, and 3 that exclude short-term incompleteness (STAI).

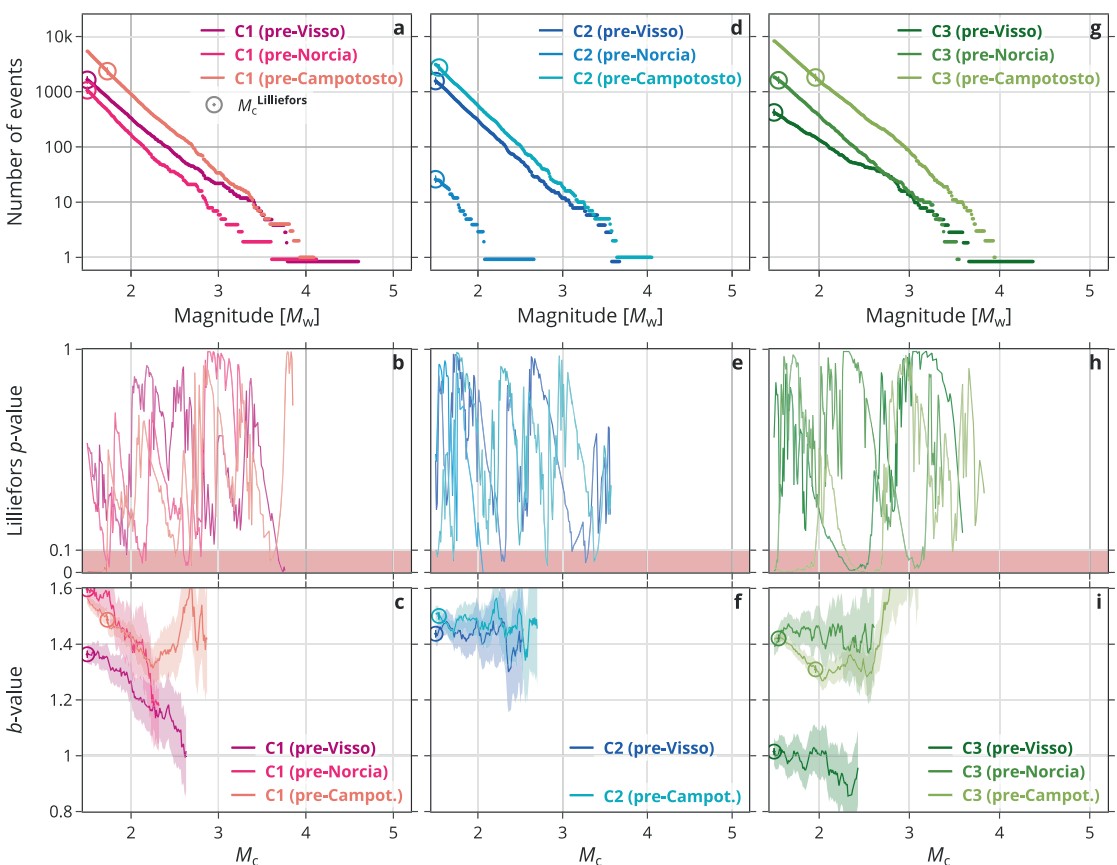

**Fig. 4 | Magnitude statistics of the three largest clusters in three individual periods. a–c** Cluster 1, **d–f** Cluster 2, and **g–i** Cluster 3. Like Fig. 3, **a**, **d**, and **g** show clusters in terms of their magnitude–frequency distribution; **b**, **e**, and **h** show the Lilliefors *p*-value as function of $M_c$; and **c**, **f**, and **i** the *b*-value as function of $M_c$ with the shading representing 1$\sigma$. Supplementary Fig. 4 shows the same analysis using local magnitudes. Supplementary Figs. 7 and 8 compare the periods shown here with periods that include STAI.

additional aspects influence and potentially bias *b*-value estimates and are not always carefully addressed. We emphasize that the absolute *b*-value has little meaning not only due to its dependence on the magnitude scale (Supplementary Note 2.3), but also on the particular conversion relation (Supplementary Note 2.5 and Supplementary Fig. 9).

Generalizing our findings, we hypothesize that a complex and distinct MFD behavior is not unique to the CI2016 sequence, but likely occurs in other regions and sequences. In particular, our results predict that in a larger region (like the extent of the CI2016 sequence), the temporal *b*-value variability must exceed the expected natural

variability (i.e., uncertainty) of a constant *b*-value, as found in various regions[5,11,48,53]. Our method may be beneficial for studying the peculiarities of spatiotemporal MFD variability to better understand the processes that influence seismicity. For instance, it may aid in exploring and modeling stress heterogeneities to improve the earthquake forecasting skill of physics-based models[63–65]. Even if the physical processes remain hidden, merely recognizing that the MFD behaves in a complex manner potentially improves the spatiotemporal forecast performance—it is a way to better appreciate the fine-scale heterogeneity and complexity of activated tectonic structures. Future work may focus on a refined identification of spatiotemporal clusters to

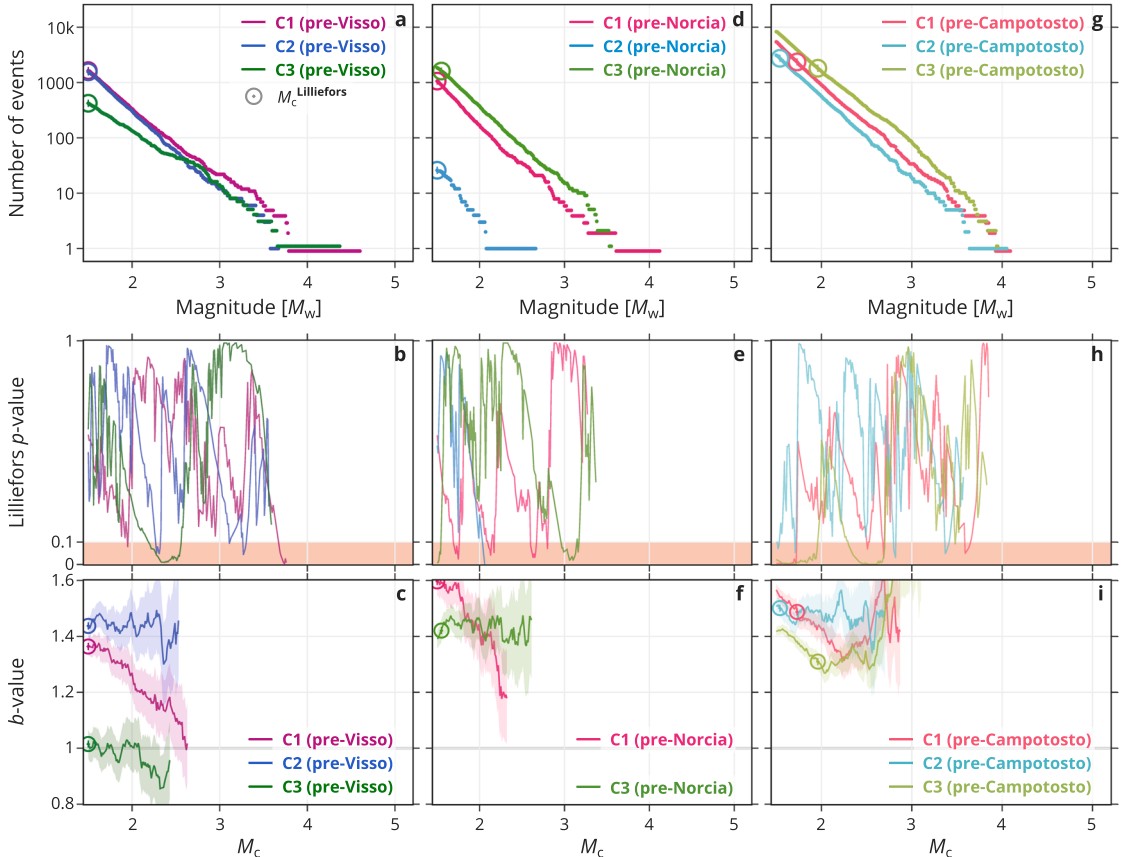

**Fig. 5 | Reordering the magnitude statistics of Fig. 4 temporally by period.**
**a**–**c** 'pre-Visso', **d**–**f** `pre-Norcia', and **g**–**i** `pre-Campotosto'. Like Figs. 3 and 4,
**a**, **d**, and **g** show clusters in terms of their magnitude–frequency distribution;
**b**, **e**, and **h** show the Lilliefors $p$-value as function of $M_c$; and **c**, **f**, and **i** the $b$-value as
function of $M_c$ with the shading representing $1\sigma$. Supplementary Fig. 5 shows the
same analysis using local magnitudes.

better connect them with individual tectonic structures, possibly by
not relying solely on hypocenter density.

## Methods

### High-resolution earthquake catalog of the sequence

We used the high-resolution catalog of Tan et al.[50], which spans from
2016-08-15 to 2017-08-15, and extracted a spatial subset as follows:
depth < 12 km; UTM easting: 330–370 km (about longitude
12.94–13.40); UTM northing: 4690–4790 km (about latitude
42.34–43.25). Only earthquakes with moment magnitudes $M_w \geq 1.5$
were considered, totaling 76 055 events. The provided hypocenters are
based on high-precision relative relocation. The $M_w$ contained in the
catalog were converted from local magnitudes, $M_L$, with an average
European scaling relation[66] based on a polynomial fit using catalogs of
different seismological agencies with most events having $M_L > 1.5$ and
$M_w \gtrsim 1.5$, but adjusted to regional data (Supplementary Note 2.3).

### Identifying spatial earthquake clusters and creating temporal subsets

To infer the spatial distribution of seismogenic zones, we followed
recommendations based on density-based clustering analyses of
earthquake catalogs[67]. Accordingly, earthquake hypocenters were
spatially separated into clusters using DBSCAN (Density-Based Spatial
Clustering of Applications with Noise)[68], which groups points based on
how closely they are packed together and allows identifying volumes
of arbitrary shapes; points that lie in low-density zones are left as
outliers. Because the horizontal extension of the CI2016 sequence is
several times larger than the vertical extension, density-connected
clouds of hypocenters preferentially extend in horizontal directions.
To improve the clustering analysis for such an anisotropic case, we

rescaled the hypocenter coordinates to a uniform extent in each
direction, i.e., rescaled into a cube. This procedure increased the local
hypocenter density in horizontal planes, which facilitated identifying
hypocenter clusters with horizontally elongated shapes[67] (Supplementary Fig. 1). DBSCAN was then applied with parameter values that
led to an optimal clustering solution[67]: $\epsilon = 0.40$, the neighborhood
radius and $Z = 200$, the minimum number of points required to form a
dense region. This configuration produced nine clusters, from which
we selected the five largest (C1–5, descending by size) and labeled the
remaining earthquakes as 'unclustered'. Their spatial distribution is
shown in Fig. 1 and the data is provided as Supplementary Data 1.

To enable a temporal analysis, each of the largest clusters C1–C3
was divided into three periods (see indicators in Fig. 2):
- early: earthquakes between the Amatrice and Visso event;
- mid: earthquakes since the Visso event until 2 days after the
  Norcia mainshock;
- late: the rest.C4 and C5 contain too few data to benefit from this
  division.

As illustrated in Supplementary Fig. 6, these periods are affected
by short-term aftershock incompleteness (STAI, see also Supplementary Note 2.4)[69–72]. Supplementary Fig. 6 makes use of equalized plot
scales[73] and overlays the point density as suggested by W. Ellsworth
(pers. comm., 2021). In this way, Supplementary Fig. 6 informs us
about the STAI duration after each main event, leading us to exclude
STAI by using temporal subsets of the three periods for C1, C2, and C3
(see indicators at the top of Fig. 2):
- pre-Visso: like 'early', but excluding the first 0.8 days after the
  Amatrice event;
- pre-Norcia: like 'mid', but excluding the first 0.6 days after the
  Visso event and 2 days after the Norcia mainshock;

- pre-Campotosto: like 'late', but before the Campotosto event;
- post-Campotosto: like 'late', but after the Campotosto event excluding the first 0.4 days.

## Earthquake statistics

The clusters and their temporal subsets were investigated in terms of their MFD. To quantify MFD variability, we calculated the b-value as function of $M_c$ for each cluster or temporal subset. The b-value was determined using a bias-free maximum likelihood estimation[22,74] for sample sizes $N \geq 50$. The b-value requires an exponential distribution of the magnitude above $M_c$ to be physically meaningful[25]. To assess the exponentiality of the MFD, we applied the Lilliefors test[25,26] using the implementation of Herrmann and Marzocchi[75] and obtain a p-value as function of $M_c$, which expresses the probability to observe the MFD assuming that the exponential distribution is the underlying distribution. For a significance level of $\alpha = 0.1$, we derived the lowest magnitude level for which the MFD can be considered exponential, referred to as $M_c^{\text{Lilliefors}}$. We always refer to the b-value at $M_c^{\text{Lilliefors}}$.

As an alternative to quantify MFD variability, we used the two-sample Kolmogorov–Smirnov (KS) test and compared the MFD of clusters or their temporal subsets pairwise. For each pair, the largest $M_c^{\text{Lilliefors}}$ was used as lower magnitude cutoff. The KS test returns a p-value as a measure for the strength of evidence against the null hypothesis that the two MFDs come from the same parent distribution. We interpreted a p-value < 0.05 as a statistically significant difference.

We assumed that distinct b-values or significant p-values reflect differences or changes of the entire exponential part of the MFD. This assumption is rather simplistic because both metrics are not ideal representations of the MFD: While the b-value estimate increasingly correlates with the largest magnitude for decreasing sample sizes[25], the KS test has a generally reduced sensitivity for differences toward the tails of the distributions. But they are the most widely accepted metrics available for representing the MFD or their differences. We did not explore whether the MFD could be characterized by a tapered GR distribution, and therefore neglected variations of the maximum magnitude, e.g., due to released energy close to faults[28].

## Consider prior seismicity?

The high-resolution catalog of Tan et al.[50] only contains 15 events with $M_w \geq 1.5$ before the first main event (the Amatrice event). To better represent prior seismicity in the MFD analyses (e.g., as a reference "background" b-value), we initially considered HORUS[76] (horus.bo.ingv.it) as a temporally extensive catalog that provides $M_w$ magnitudes. HORUS has already been used to study the CI2016 sequence in terms of b-value[48]. The $M_w$ in HORUS were converted from $M_L$ with a different scaling relation[77] than the $M_w$ in the high-resolution catalog[66]. (Note that the $M_L$ estimates in either catalog are also not based on the same procedure.) In fact, a comparative MFD analysis for CI2016 seismicity shows that the b-value differs considerably between both catalogs (0.2 units at $M_c^{\text{Lilliefors}}$, see Supplementary Note 2.5 and Supplementary Fig. 9). The two $M_w$ scales are not consistent with each other, rendering a reliable MFD comparison impossible. We therefore did not use the HORUS catalog and omitted prior seismicity in our MFD analyses.

## Data availability

The high-resolution earthquake catalog of Tan et al.[50] is available at Tan et al.[78]. The extracted events with their associated cluster indices obtained in this study can be found in Supplementary Data 1.

## Code availability

No proprietary code was used in this study; the performed analysis can be reproduced from the specifications in the Methods section. The density-based cluster identification following the recommendations of Piegari et al.[67] is based on the DBSCAN algorithm implemented in Matlab (www.mathworks.com/help/stats/dbscan.html); $M_c^{\text{Lilliefors}}$ was calculated with the Python class of Herrmann and Marzocchi[75]. Figures were created with the Python graphing library plotly (www.plotly.com/python).

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

## Acknowledgements

This study was supported by the *'Real-time Earthquake Risk Reduction for a Resilient Europe'* (RISE) project, funded by the European Union's *Horizon 2020* research and innovation program (Grant Agreement No. 821115).

## Author contributions

M.H. performed the statistical analyses, created the figures, and wrote the manuscript. E.P. performed the clustering analysis and reviewed the manuscript. W.M. outlined and led the project and reviewed the manuscript. All authors designed the study and discussed the results.

## Competing interests

The authors declare no competing interests.
