## [Peer Review File · Nature Communications]

Revealing the spatiotemporal complexity of the magnitude distribution and b -value during an earthquake sequenceREVIEWER COMMENTS

Reviewer #2 (Remarks to the Author):

The manuscript of Herrmann et al. delivers exciting new insight on the variation of the frequency-magnitude distribution of earthquakes within a sequence. The focus of the works is on a very hot topic of research, whether b -value can be informative of impending hazard given the fact that its variability and controls are not well-understood. The manuscript is clearly an essential contribution to the field of earthquake forecasting, and its results are scientifically sound and the method is reproducible. I have no doubt that the manuscript should be published and will be well-read. I have only few minor suggestions (as clarifications) but it could very well be that the paper can be published in its present form.

Ln 197-198, page 10 "in the structure where", perhaps use another term instead of structure that suggests that the rupture surface, or identified plane related to the rupture, was used to infer clusters. Could be "in the locality".

Ln 207-208, "became insensitive to strong seismicity", please add a comment about the location of the after-Norcia, Campotosto events. Their hypocenters at the south part of the sequence, far from Clusters 1-3. Please clarify if by "strong seismicity" you mean the Campotosto events.

Ln 227-229, Recent work on the development of physics-based forecasts also supports the findings of the Authors here about the importance of heterogeneity in understanding earthquake occurrence and improving forecasting. Segou and Parsons (2020) show that earthquake occurrences are related with the heterogeneity of the pre-existing stress field, Mancini et al. (2019; 2020) show that incorporating stress-field and fault heterogeneity as the earthquake sequence evolves is the key to improve model predictability. Hardebeck (2020), cites the above work, while showing the importance of incorporating information about the stress field heterogeneity in statistically-derived earthquake probabilities. Considering the above, the Authors are right to assume that heterogeneity may control frequency-magnitude statistics.

Ln 201-202 Please provide a reference for the sentence "The generally...accumulate stress"

Ln 241-243, Please provide a reference for the sentence "(ii) a consequence..." and, perhaps consider rephrasing towards the fact that in the large-scale we may see that aftershocks generally reduce differential stress in the long-term, but the secondary triggering effects on isolated planes indicate that their (aftershocks) contribution is important in local stress-field evolution.

Reviewer #3 (Remarks to the Author):

The paper investigates Magnitude-Frequency-Distribution (MFD) for the seismicity pattern of the central Italy 2016 sequence (CI2016). The MFD is modeled with the GR relation to retrieve the b -value, the parameter controlling the relative rate of small and large earthquakes.

The work deals with a rigorous estimation of the impact of key elements such as quality of the input data (earthquakes catalogue), its space-time reckoning, the magnitude scale and completeness over time, in relation to the calculation of the b -value.

Looking at b -value in both space and time may help to understand the earthquake occurrence process and positively contribute to earthquake forecasting.

By means of the selection of clusters within a high-resolution catalogue describing the CI2016, the Authors observe "unexpected" spatiotemporal MFD variability, including b -value increases in the cluster where the largest event of the sequence will occur.

To explain this behavior, the Authors suggest for an effect of the heterogeneity and complexity of the tectonic structure on the MFD.

In my opinion the paper is timely and definitively suitable for publication on NC. The analysis is rigorous, and the topic is a key one for evaluating the possibility to use b-value in short-term seismic hazard assessment. At the same time there are important aspects to be clarified before its publication.

TETCONIC

To investigate the space time pattern of MFD and b-value, the Authors identify, and isolate clusters of seismicity based on hypocenter density. As stated for the larger cluster (C1; lines 124-126) and as well as visible in Figure 1, the clusters are not sampling individual tectonic structures such as specific fault segments and detachments.

With this approach I believe that Authors are not accounting for the "complexity of the tectonic structures" (e.g., line 20 of the Abstract). That is why I recommend modifying the claims for the effect of heterogeneous stress field and complex fault geometry in generating MFD variability (lines 69-75, lines 114-116 and 225-247), while posing the attention on the intrinsic difficulty in a real time and non-arbitrary (specific, meaningful, and reproducible – GW21 and DCetal21) selection of an appropriate spatiotemporal scale for resolving b-value in space and time. I think this is the key aspect that the work demonstrates exceptionally well, thus posing doubts on the possibility to use this parameter in a forecasting setting. In my view, the connection between the investigated parameters and the geological faults has not been explored.

The rescaling of the hypocenter coordinates to improve the clustering analysis (described between lines 288-294) is adequate but I strongly recommend providing Figure 1 without vertical exaggeration. This is a key point allowing the reader to understand the relationship between faults geometries highlighted by seismicity distribution and clusters.

MAGNITUDE SCALE

To gain a longer (through more magnitudes classes) exponential distribution, the availability of a catalogue possessing moment magnitude for each earthquake would be preferable. But if I well understood, Tan et al., 2021 provide two magnitudes for each event in their catalogue: a computed local magnitude (ML) and a moment magnitude (MW) derived from the relation specified by Gruntal et al., 2009. As also declared by the Authors, the high b-values they observe are undoubtedly due to a consistent (and implicit) overestimation of the smaller magnitudes when passing from ML to MW. However, the used relation is not tailored for the area.

On the same subject, when using the MW, the MCLillieffors of the entire catalogue is significantly higher than the ones of all the clusters (see Figure 3 top panel). On the contrary, using the ML the MCLillieffors of the entire catalogue is (at least) similar to the one of the largest cluster (C1; see Figure S3 top panel). I kindly ask to the Authors to comment on this.

Nevertheless, when using ML for the individual periods, the discrepancies generated by the two classes of magnitudes are reduced and the main findings remain unaffected. My suggestion is to bring within the main text of the paper the key points of the discussion about the influence of magnitude scale. While I suggest removing all the discussion and comparison made with the HORUS catalogue and anthropogenic signals that do not provide any additional information to the key points of the papers, apart arising additional qualms on the use of earthquakes catalogues again with moment magnitudes derived from regressions.

GENERAL COMMENTS

Adding a figure with the resolved b-values versus time (including mainshocks occurrence time) would be important to provide a synthesis about all cluster's behavior. Instead of showing only the number of events versus magnitude for the diverse periods.

Why in the top panel of Figure 3, C1 doesn't reach MW6.5, as for the magnitude of the Norcia mainshock contained in this cluster?

I suggest openly discussing the implications of the retrieved increase of the b-value in the cluster (C1) containing the Norcia hypocenter (nucleation point), before its occurrence, with respect to Gulia and Wiemer, 2021 and Garcia-Hernandez et al., 202. This is a significant point.

Please, add a comment on the choice of the DBSCAN clustering parameters.

I suggest adding in the main figures (e.g., Figure 1) details about the plotted catalogue.

In Figure 4 and 5 (plus corresponding figures in the Supplementary) I suggest using an easier way to identify the clusters (e.g., continuous, and dashed lines, points etc.). The reason for using the same colors identifying the clusters is a good one, but the printed version of the manuscript is not good enough.

Abstract

Line 15. I would avoid using "unexpected" here.

Line 20. I would change "tectonic structures" with "system".

We are pleased to resubmit a revised version of NCOMMS-21-51551 “*b-value of what? Spatiotemporal complexity of the magnitude distribution during the 2016–2017 central Italy sequence*” (note the slightly changed title). We express our sincere thanks to the two reviewers for their remarks, suggestions, and corrections. The manuscript has been considerably improved by addressing their feedback. You find detailed responses to the reviewers' concerns below.

All changes are documented in the annotated manuscript. The largest change related to the first comment of Reviewer #3 (clarifying that we are not too specific about the obtained tectonic structures). We also used this revision as opportunity to further improve the clarity and wording of our statements throughout the manuscript. Almost all sentences that made use of in-text citations were adjusted to using superscript-style citation. In the Introduction, we added another reference published in the meantime that relates to “*recent scientific studies [that] reemphasized that choices have to be specific, meaningful, and reproducible*”: DeSalvio & Rudolph (2021). In addition, we added a ‘Code Availability’ section.

DeSalvio, N. D., & Rudolph, M. L. (2022). A retrospective analysis of b -value changes preceding strong earthquakes. *Seismological Research Letters*, 93(1), 364–375. doi: [10.1785/0220210149](https://doi.org/10.1785/0220210149)

REVIEWER #2 COMMENTS:

The manuscript of Herrmann et al. delivers exciting new insight on the variation of the frequency-magnitude distribution of earthquakes within a sequence. The focus of the works is on a very hot topic of research, whether b -value can be informative of impending hazard given the fact that its variability and controls are not well-understood. The manuscript is clearly an essential contribution to the field of earthquake forecasting, and its results are scientifically sound and the method is reproducible. I have no doubt that the manuscript should be published and will be well-read. I have only few minor suggestions (as clarifications) but it could very well be that the paper can be published in its present form.

We appreciate the reviewer’s positive comment and respond to his/her minor comments below.

Ln 197-198, page 10 “in the structure where”, perhaps use another term instead of structure that suggests that the rupture surface, or identified plane related to the rupture, was used to infer clusters. Could be “in the locality”.

We replaced ‘structure’ with ‘cluster’ to name the correct spatial reference.

Ln 207-208, “became insensitive to strong seismicity”, please add a comment about the location of the after-Norcia, Campotosto events. Their hypocenters at the south part of the sequence, far from Clusters 1-3. Please clarify if by “strong seismicity” you mean the Campotosto events.

We added “, which occurred in the southern part of the sequence” to the previous sentence to inform about their location. Note that the Campotosto events indeed occur farther away from C1 & C3, but close to C2, which also remains unchanged in terms of b -value.

We replaced “*strong seismicity*” with “*large earthquakes*” to indicate that eventually any notable event did not influence the MFD in C1-C3 further (note that this change reuses the same term as in the statement “*large earthquakes may influence the MFD*”

a few lines earlier and better constraints it).

Ln 227-229, Recent work on the development of physics-based forecasts also supports the findings of the Authors here about the importance of heterogeneity in understanding earthquake occurrence and improving forecasting. Segou and Parsons (2020) show that earthquake occurrences are related with the heterogeneity of the pre-existing stress field, Mancini et al. (2019; 2020) show that incorporating stress-field and fault heterogeneity as the earthquake sequence evolves is the key to improve model predictability. Hardebeck (2020), cites the above work, while showing the importance of incorporating information about the stress field heterogeneity in statistically-derived earthquake probabilities. Considering the above, the Authors are right to assume that heterogeneity may control frequency-magnitude statistics.

We very much appreciate this valuable context of recent studies about the role of stress heterogeneity. We consider these references very valuable and incorporated them as follows in the last part of the Discussion on LINE 268 FF (after “*Our method may be beneficial for studying the peculiarities of spatiotemporal MFD variability to better understand the processes that influence seismicity.*”):

“For instance, it may aid in exploring and modeling to improve the earthquake forecasting skill of physics-based models [Segou and Parsons 2020, Mancini et al. 2020, Hardebeck 2021].”

Ln 201-202 Please provide a reference for the sentence “The generally...accumulate stress”

We prepended the following statement to the beginning of this sentence: “*With the b-value being inversely related to the differential stress [Wyss 1973; Scholz 2015], ...*” and removed “(i.e., low differential stress)” at the end of it.

Ln 241-243, Please provide a reference for the sentence “(ii) a consequence...” and, perhaps consider rephrasing towards the fact that in the large-scale we may see that aftershocks generally reduce differential stress in the long-term, but the secondary triggering effects on isolated planes indicate that their (aftershocks) contribution is important in local stress-field evolution.

We reshaped this point as follows for clarity: “(ii) *the previous two main events (Amatrice and Visso) and their aftershocks gradually releasing built-up strain and reducing the overall stress in the fault zone.*”. That this aspect relates to the b-value should be clearer by having addressed the reviewer’s previous comment (i.e., b-value being related to the stress state); we did not consider it necessary to add a reference about the fact that earthquakes release strain and cause stress drops. The second aspect (i.e., secondary aftershocks shaping the local stress field) sounds interesting, but we did not include it (i) due to its minor relevance in explaining the observed b-value behavior and (ii) to avoid confusion.

REVIEWER #3 COMMENTS:

In my opinion the paper is timely and definitively suitable for publication on NC. The analysis is rigorous, and the topic is a key one for evaluating the possibility to use b-value in short-term seismic hazard assessment. At the same time there are important aspects to be clarified before its publication.

We also appreciate this reviewer’s positive opinion and respond to his/her minor comments below.

TECTONIC:

To investigate the space time pattern of MFD and b-value, the Authors identify, and isolate clusters of seismicity based on hypocenter density. As stated for the larger cluster (C1; lines 124-126) and as well as visible in Figure 1, the clusters are not sampling individual tectonic structures such as specific fault segments and detachments.

With this approach I believe that Authors are not accounting for the “complexity of the tectonic structures” (e.g., line 20 of the Abstract). That is why I recommend modifying the claims for the effect of heterogeneous stress field and complex fault geometry in generating MFD variability (lines 69-75, lines 114-116 and 225-247), while posing the attention on the intrinsic difficulty in a real time and non-arbitrary (specific, meaningful, and reproducible – GW21 and DCetal21) selection of an appropriate spatiotemporal scale for resolving b-value in space and time.

I think this is the key aspect that the work demonstrates exceptionally well, thus posing doubts on the possibility to use this parameter in a forecasting setting. In my view, the connection between the investigated parameters and the geological faults has not been explored.

The addressed limitation deserves to be explained better. It is true that a seismic sequence (almost) never illuminates (i.e., that a seismic catalog never samples) the tectonic structures entirely (and therefore the full range of complexity), but rather only the parts that are active at that moment. We indeed do not want to appear too confident and deterministic about which tectonic structures we have isolated of the fault system. Such a specific connection of the most active volumes (and their MFD) with the tectonic and structural setting has to be explored better in future works. Yet, our observation that the MFD varies should not overly depend on how the fault system was spatiotemporally sampled: If the b-value were constant and independent from the geological structures, we would never obtain the distinct variations that we observed in our study, no matter how the seismicity is (randomly) sampled in space and time. In essence, we still speculate that the observed (complex) variation of the MFD reflects a complex and heterogeneous fault system (and stress field); an aspect Reviewer #2 agreed with.

We have modified the text to improve this matter:

1. We clarified that we do not isolate individual tectonic structures by replacing several terms and statements as follows:
 - ABSTRACT: “*strong influence of the heterogeneity and complexity of tectonic structures on the MFD*”: “*the fault system's heterogeneity and complexity influence the MFD*”;
 - LINE 70: “*tectonic structures*”: “*distinct zones of a fault system*”;
 - LINE 73: “*the internal structure*”: “*distinct seismogenic zones*”;
 - LINE 112: “*account for the complexity of the sequence including its depth-dependent structure*”: “*use a specific spatiotemporal scale that considers the three-dimensional distribution of seismicity*”;
 - LINE 115: “*complex sequence in an isolated fashion*”: “*distribution of the most seismogenic zones*”;
 - LINE 201: “*same structures*”: “*same seismogenic zones*”;
 - LINE 250: “*observed substructure*”: “*observed seismogenic zone*”;
 - LINE 251: “*most active structures*”: “*most seismogenic zones*”;
 - LINE 252: “*individual structures*”: “*individual clusters*”;

- LINE 255: “*individual structures*”: “*individual seismogenic zones*”.
2. We restructured the Discussion to separate the focus and main implication of our study (appropriate spatiotemporal scale for resolving b-values) from our possible interpretations and speculations (heterogeneous stress-field, complex structure). In this way, we put more emphasis on our main intent without neglecting possible interpretations and future directions:
- LINE 193 f. (at the beginning of the Discussion): Replaced “*This observed MFD variability is likely due to the fine-scale heterogeneity and complexity of the tectonic structures that were activated in this sequence.*” with “*Finding this complex MFD variability highlights that the spatiotemporal scale to select seismicity and resolve the b-value has to be specific and meaningful lacking reproducibility of related studies and facilitates the unified interpretation of MFD and b-value variability.*”;
 - LINE 229 f.: We added a description for our main speculation: “*In other words, the different spatiotemporal behavior of the elastic energy release may be related to some particular (and unknown) features of the geological setting.*”;
 - LINE 258 f.: Moved the following statement from LINE 271 f. (formerly penultimate sentence): “*... reduce the amount of expert judgement and subjective choices in MFD analysis.*”;
 - LINE 260 f.: Added the statement “*For instance, a spatial scale based on density-based clustering has the potential to provide appropriate reference [...] volumes paving the way for reproducible and replicable results regarding b-value estimates.*”;
 - LINE 260 ff.: We separated the following text into a new paragraph starting with “*Generalizing our findings, ...*”;
 - LINE 268 ff.: Note that we extended our discussion with the role of the stress heterogeneity on earthquake occurrence and forecasting, based on the third comment of Reviewer #2.
 - LINE 270 f.: Added the following statement to the now-penultimate sentence: “*—it is a way to better appreciate the fine-scale heterogeneity and complexity of activated tectonic structures.*”
 - LINE 273 f.: Replaced “*improve MFD and b-value analysis*” with “*better connect them with individual tectonic structure*“ in the last sentence as outlook for future work.

The rescaling of the hypocenter coordinates to improve the clustering analysis (described between lines 288-294) is adequate but I strongly recommend providing Figure 1 without vertical exaggeration. This is a key point allowing the reader to understand the relationship between faults geometries highlighted by seismicity distribution and clusters.

We followed this advice and provide unscaled depth slices.

MAGNITUDE SCALE:

To gain a longer (through more magnitudes classes) exponential distribution, the availability of a catalogue possessing moment magnitude for each earthquake would be preferable. But if I well understood, Tan et al., 2021 provide two magnitudes for each event in their catalogue: a computed local magnitude (ML) and a moment magnitude (MW) derived from the relation specified by Gruntal et al., 2009. As also declared by the Authors, the high b-values they

observe are undoubtedly due to a consistent (and implicit) overestimation of the smaller magnitudes when passing from ML to MW. However, the used relation is not tailored for the area.

The magnitude conversion relation may not be ideal, but we went with the simplest solution: taking the original M_w estimates provided with the catalog. Alternatives are the relation used in *Lolli et al. (2020)*, i.e., the HORUS catalog, a revision of *Gasperini et al., 2013*, or the bilinear relation of *Malagnini and Munafò (2018)*; but using either of those would increase the entropy of our analysis and inevitably the “influence of expert choices” as stated in our manuscript—a condition that we want to avoid as much as possible as advocated by this very manuscript. In this sense, the original catalog creators did the choice for us. The Grünthal relation is not unsuitable: it is based on European data, which includes data of Italy (1983–2004). *Tan et al. (2021)* did not use this relation directly, but calibrated the constant (using ~500 events with M_w estimated from regional waveform fitting [*Herrmann et al. 2011*]), as specified in their supplemental material. We originally mentioned the exact adjustment in our manuscript, but removed this detail; we simply added “*but adjusted to regional data*”, but included the whole argumentation of this answer in Supplementary Note 2.3 (*Influence of magnitude scale*).

Lolli, B., D. Randazzo, G. Vannucci & Gasperini, P. (2020). The homogenized instrumental seismic catalog (Horus) of Italy from 1960 to present. *Seismological Research Letters*, 91(6), 3208–3222. doi: [10.1785/0220200148](https://doi.org/10.1785/0220200148)

Gasparini, P., B. Lolli & G. Vannucci (2013). Empirical calibration of local magnitude data sets versus moment magnitude in Italy. *Bulletin of the Seismological Society of America*, 103(4), 2227–2246. doi: [10.1785/0120120356](https://doi.org/10.1785/0120120356)

Malagnini L. & I. Munafò (2018). On the Relationship between ML and M_w in a Broad Range: An Example from the Apennines, Italy. *Bulletin of the Seismological Society of America*, 108(2), 1018–1024. doi: [10.1785/0120170303](https://doi.org/10.1785/0120170303)

Herrmann, R. B., L. Malagnini, & I. Munafò (2011). Regional moment tensors of the 2009 L'Aquila earthquake sequence, *Bulletin of the Seismological Society of America*, 101, 975–993.

On the same subject, when using the MW, the MCLillieffors of the entire catalogue is significantly higher than the ones of all the clusters (see Figure 3 top panel). On the contrary, using the ML the MCLillieffors of the entire catalogue is (at least) similar to the one of the largest cluster (C1; see Figure S3 top panel). I kindly ask to the Authors to comment on this.

The $M_c^{\text{Lillieffors}}$ using M_w is higher when using all data due to the included incompleteness. Using M_L additionally includes the scaling break of the local magnitude, as mentioned in Supplementary Note 2.3; for more details, please have a look at *Herrmann & Marzocchi (2021)*. Note also the M_L -based b -value continuously increases with cutoff magnitude, M_c .

Herrmann, M., & W. Marzocchi (2021). “Inconsistencies and lurking pitfalls in the magnitude–frequency distribution of high-resolution earthquake catalogs. *Seismological Research Letters*, 92(2A), 909–922. doi: [10.1785/0220200337](https://doi.org/10.1785/0220200337)

Nevertheless, when using ML for the individual periods, the discrepancies generated by the two classes of magnitudes are reduced and the main findings remain unaffected. My suggestion is to bring within the main text of the paper the key points of the discussion about the influence of magnitude scale. While I suggest removing all the discussion and comparison made with the HORUS catalogue and anthropogenic signals that do not provide any additional information to the key points of the papers, apart arising additional qualms on the use of earthquakes catalogues again with moment magnitudes derived from regressions.

The influence of the magnitude scale on the MFD and b-value has been exhaustively addressed by our previous study [Herrmann & Marzocchi 2021] and additionally in the supplement as one source of b-value influence and bias. We strongly believe that only the M_w should be used for MFD and b-value analysis; the ML-based results are merely “for the sake of completeness”, to highlight that it does not always lead to the same results. For these reasons, we do not want to highlight this aspect further in the main text.

We agree that the subsection in the method section about the HORUS catalog does not affect our key findings, but we consider it important to mention that our obtained b-values are different from those based on the HORUS catalog (e.g., note that Gulia & Wiemer (2019) based their analysis of the same sequence on the HORUS catalog). We have renamed the subsection “Prior seismicity” with “Consider prior seismicity?” and added a reference to Gulia & Wiemer (2019).

We did not refer to a possible contamination of anthropogenic signals in the main text, but included this aspect in the supplement merely due to a recent discussion about the influence of quarry blasts on the MFD analysis; this discussion has now been published [Gulia et al. 2022; Taroni et al. 2022]. We added those references to the supplement.

Gulia, L., Gasperini, P., & S. Wiemer (2022). Comment on “High-definition mapping of the Gutenberg–Richter b -value and its relevance: A case study in Italy” by M. Taroni, J. Zhuang, and W. Marzocchi. *Seismological Research Letters*, 93(2A), 1089–1094. doi: [10.1785/0220210159](https://doi.org/10.1785/0220210159)

Taroni, M., J. Zhuang, & W. Marzocchi (2022). Reply to “Comment on ‘High-definition mapping of the Gutenberg–Richter b-value and its relevance: A case study in Italy’ by M. Taroni, J. Zhuang, and W. Marzocchi” by Laura Gulia, Paolo Gasperini, and Stefan Wiemer. *Seismological Research Letters*, 93(2A), 1095–1097. doi: [10.1785/0220210244](https://doi.org/10.1785/0220210244)

GENERAL COMMENTS:

Adding a figure with the resolved b-values versus time (including mainshocks occurrence time) would be important to provide a synthesis about all cluster’s behavior Instead of showing only the number of events versus magnitude for the diverse periods.

We understand that a temporal resolution of b-values is a desired representation of our analysis. However, the limited sample size of most clusters would not allow for a robust sliding-window-based b-value estimation. Instead, to guarantee robust estimates, we looked at the MFD period-wise.

We also did not want to provide an overview plot of those few b-value estimates, because, as argued in the introduction, we did not only want to focus solely on the b-value variation, but the evolution of the MFD itself (incl. the dependence of the b-value as function of M_c). For this reason, we have considered it more useful to show Figure 4 reordered in time (Figure 5).

Why in the top panel of Figure 3, C1 doesn’t reach $M_w6.5$, as for the magnitude of the Norcia mainshock contained in this cluster?

The $M_w6.5$ refers to the Time Domain Moment Tensor (TDMT) estimate as provided by the INGV bulletin (<http://terremoti.ingv.it/event/8863681>). We used the estimate provided by Tan et al. (2021), $M_w6.06$ (see also Fig. 2), based on an M_L estimate converted to M_w with the Grünthal relation. As mentioned above, other sets of M_w estimates for events of the CI2016 sequence exist:

- *Malagnini and Munafò* (2018) estimated a M_w 6.33 for the Norcia event. This set is also propagated by other catalogs of the CI2016 sequence [e.g., *Spallarossa et al.* 2020; *Waldhauser et al.* 2021];
- *Lolli et al.* (2020, i.e., the HORUS catalog) estimated a M_w 6.61.

Spallarossa D., M. Cattaneo, D. Scafidi, M. Michele, L. Chiaraluca, M. Segou and I. G. Main (2021). An automatically generated high-resolution earthquake catalogue for the 2016-2017 Central Italy seismic sequence, including P and S phase arrival times. *Geophysical Journal International*, 225, 555–571. doi: 10.1093/gji/ggaa604

Waldhauser, F., M. Michele, L. Chiaraluca, R. Di Stefano, and D. P. Schaff (2021). Fault Planes, Fault Zone Structure and Detachment Fragmentation Resolved with High-Precision Aftershock Locations of the 2016–2017 Central Italy Sequence. *Geophysical Research Letters*, 48(16), 1–10. doi: 10.1029/2021GL092918.

I suggest openly discussing the implications of the retrieved increase of the b-value in the cluster (C1) containing the Norcia hypocenter (nucleation point), before its occurrence, with respect to Gulia and Wiemer, 2021 and Garcia-Hernandez et al., 202. This is a significant point.

We believe that we sufficiently referred to this aspect in the Discussion section, including possible reasons for its origin. At present we offer no further explanations for this observation.

Please, add a comment on the choice of the DBSCAN clustering parameters.

We now refer to a paper we were working in parallel to this study: *Piegari et al.* (2022), which was published just recently. It is supposed to serve as a guideline for the cluster analysis of seismic catalog using density-based clustering algorithms; in that study, we explored the parameter space exhaustively and analyzed the associated cluster solutions. We added this sentence at the beginning of this paragraph:

“To infer the spatial distribution of seismogenic zones, we followed recommendations based on density-based clustering analyses of seismic catalogs [Piegari et al. 2022]. Accordingly, ...”

and replaced *“We then applied DBSCAN with the following parameters:”* with:

“DBSCAN was then applied with parameter values that led to an optimal clustering solution [Piegari et al. 2022]:”

Piegari, E., M. Herrmann, W. Marzocchi (2022). 3D spatial cluster analysis of seismic sequences through density-based algorithms, *Geophysical Journal International*, 2022, ggac160, doi: [10.1093/gji/ggac160](https://doi.org/10.1093/gji/ggac160)

I suggest adding in the main figures (e.g., Figure 1) details about the plotted catalogue.

Not clear which details the Reviewer is referring to. In any case, we provide the catalog in the supplement (and how the subset was extracted in the text), which allows anyone to obtain any desired detail.

But we now added an overview map as an inset to indicate the location of the shown area in Italy.

In Figure 4 and 5 (plus corresponding figures in the Supplementary) I suggest using an easier way to identify the clusters (e.g., continuous, and dashed lines, points etc.). The reason for using the same colors identifying the clusters is a good one, but the printed version of the manuscript is not good enough.

Dashed or dotted lines are more difficult to discern, however, we updated the color scheme to facilitate the distinction of the individual clusters.

ABSTRACT:

Line 15. I would avoid using “unexpected” here.

We replaced it with “remarkable”.

Line 20. I would change “tectonic structures” with “system”.

We replaced this occurrence “*distinct zones of a fault system*”, as already addressed in our answer to Reviewer #3’s first comment.

REVIEWERS' COMMENTS

Reviewer #3 (Remarks to the Author):

I reviewed the revised version of the manuscript and I can affirm that the important portions of the text I commented on have been modified following some of the suggestions I provided.

The Authors smoothed sentences related to the connection between identified clusters and geological structures.

They also better separated the outcome of the analysis with respect to the interpretations.

I still have some hesitations about the adoption of magnitudes derived by means of a regression, but I believe that now the reader can more easily form their own opinion about this important piece of work.

Thus, I suggest publishing the paper in its present form.